# Clinical Characteristics and Prognostic Factors Affecting Clinical Outcomes in Cytomegalovirus Retinitis Following Allogeneic Hematopoietic Stem Cell Transplantation

**DOI:** 10.3390/biomedicines13010242

**Published:** 2025-01-20

**Authors:** Qiaozhu Zeng, Yuou Yao, Jing Hou, Heng Miao

**Affiliations:** 1Department of Ophthalmology & Clinical Center of Optometry, Peking University People’s Hospital, Beijing 100044, China; zengqiaozhupku@163.com (Q.Z.);; 2Eye Diseases and Optometry Institute, Peking University People’s Hospital, Beijing 100044, China; 3Beijing Key Laboratory of Diagnosis and Therapy of Retinal and Choroid Diseases, Beijing 100044, China; 4College of Optometry, Peking University Health Science Center, Beijing 100191, China

**Keywords:** prognostic factor, cytomegalovirus retinitis, allogeneic hematopoietic stem cell transplantation

## Abstract

**Background/Objectives**: This study aimed to evaluate the clinical characteristics and identify the prognostic factors affecting visual outcomes, retinal detachment, and recurrence in cytomegalovirus retinitis (CMVR) patients following allogeneic hematopoietic stem cell transplantation (allo-HSCT). **Methods**: A retrospective analysis of 54 CMVR patients (84 eyes) who underwent allo-HSCT between 2015 and 2024 was conducted. Ophthalmologic and systemic evaluations were performed. The visual outcomes were classified as improvement, stabilization, and deterioration. Logistic regression and LASSO regression models were used to identify the prognostic factors. **Results**: Improved or stabilized visual outcomes were found in 22 eyes, while 62 eyes suffered from deterioration. Larger lesion areas were independently associated with poorer visual outcomes (OR 0.989, *p* = 0.002). Eight (9.5%) eyes had rhegmatogenous retinal detachment and thirteen (15.5%) eyes suffered from recurrence. Retinal detachment was significantly predicted by higher baseline aqueous CMV DNA load (OR 5.087, *p* = 0.026). Macula involvement (OR 5.322, *p* = 0.032) and more intravitreal injections (IVs) (OR 1.263, *p* = 0.008) were independent risk factors for recurrence. No systemic factors were found to be associated with the clinical outcome of eyes with CMVR. **Conclusions**: Ocular characteristics, rather than systemic factors, were more useful to predict the clinical outcome of eyes with CMVR. Routine ophthalmic screening and early intervention are essential to improving outcomes in this vulnerable population.

## 1. Introduction

Cytomegalovirus retinitis (CMVR) is a severe retinal disease typically observed in immunocompromised patients, such as in allogeneic hematopoietic stem cell transplantation (allo-HSCT) patients [1,2], which could lead to a poor visual outcome. It manifested variably according to different degrees of CMV-specific leukocytes deficit and immunosuppression [3]. Major retinal lesions include irregular white necrotic foci or edemas surrounded by granular infiltrates, spreading centrifugally along the vessels, with or without focal retinal hemorrhage [4,5,6].

Allo-HSCT is a well-established treatment for several hematological diseases. However, the broadening of indication, new preconditioning regimens, more aggressive treatment of graft-versus-host disease (GVHD), and the prolonged survivor rate after HSCT unfavorably result in more opportunistic infections, including CMVR [6,7], of which the incidence ranges from 0.2 to 5.6% [6,7,8,9]. Latent CMV activation or exogenous CMV entering the retina through the blood–eye barrier could cause CMVR. The CMV serostatus of the donor or recipient, antecedent CMV reactivation, delayed lymphocyte engraftment, and occurrence of GVHD have been proposed as risk factors for CMVR after allo-HSCT [9]. It can cause blurred vision, dark shadow floating, visual field loss, vision loss, and other severe lesions. Without in-time diagnosis and effective treatment, patients may suffer progressive retinal necrosis and optic atrophy, and eventually permanent vision loss.

The prognostic factors of CMVR after allo-HSCT have not yet been fully and intensively elucidated. Previous studies proposed that mismatched receipts may be potential negative factors affecting the occurrence of CMVR and a high intraocular CMV load at CMVR diagnosis may predict poorer visual outcomes in CMVR patients following allo-HSCT. Other studies proposed that higher visual acuity (VA) upon a diagnosis of CMVR was related to greater improvement of VA, but concurrent CMV diseases and foveal involvement were associated with poorer visual acuity. In these studies, factors associated with other ocular clinical outcomes, such as CMVR recurrence and retinal detachment, were not explored. Herein, we aimed to analyze the pattern of CMVR patients and explore the prognostic factors, both ocular and systemic, that may affect the visual outcome, rate of retinal detachment, and recurrence in patients after allo-HSCT.

## 2. Materials and Methods

### 2.1. Patients

We performed a retrospective cohort study of consecutive CMVR patients who received allo-HSCT at Peking University People’s Hospital in China between July 2015 and December 2023. The study was conducted in accordance with the Declaration of Helsinki and approved by the Peking University People’s Hospital Institutional Review Board (No. 2018PHB196-01).

CMVR was diagnosed with typical ophthalmological signs (white necrotic retina with focal hemorrhage in a fan-shaped distribution spreading centrifugally along vascular arcades and vascular sheathing), judged by a fundus specialist through indirect ophthalmoscopy with fully dilated pupils, and was confirmed by the detection of CMV DNA in the aqueous humor using a polymerase chain reaction (PCR) (>1 × 10^3^ copies/mL) [10]. The inclusion criteria were (1) the diagnosis of CMVR after allo-HSCT, (2) no other pathogens which cause similar ocular presentations to CMV, and (3) a negative serum human immunodeficiency virus (HIV) antibody. Exclusion criteria were signs of other fundus diseases; incomplete data of ocular examinations due to a poor systemic condition or other reasons; and a follow-up period of less than 1 month after CMVR diagnosis.

### 2.2. Ocular and Systemic Evaluation

At the initial and following visits, all participants underwent a comprehensive ophthalmological examination. The best-corrected visual acuity (BCVA), Goldmann intraocular pressure (IOP), inflammation in the anterior chamber/vitreous, and macula involvement were recorded at each visit. Patients also received ultrawide fundus photography (Optos Panoramic Ophthalmoscope P200MAAF 200Tx [Optos PLC, Dunfermline, Scotland]) and optical coherent tomography (OCT) (ZEISS CIRRUS HD-OCT4000 [Zeiss Meditec. Inc., Jena, Germany]). The CMVR lesion area was calculated using software Image-pro plus 6.0 (Media Cybernetics, Rockville, MD, USA), and expressed as disc area (DA).

The baseline systemic characteristics included age, gender, primary hematological disease, human leukocyte antigen (HLA) locus match, ABO match, poor graft function, GVHD, posttransplant lymphoproliferative disorders, duration of engraftment (defined as granulocyte survival), duration of lymphopenia (defined as peripheral lymphocyte <1.0 × 10^9^/L) after allo-HSCT, duration from allo-HSCT to CMVR diagnosis, cumulative duration of CMV DNAemia from allo-HSCT to CMVR diagnosis, and peak serum CMV DNA load after allo-HSCT. Additionally, the DNA levels of CMV and the Epstein–Barr virus (EBV) in the peripheral venous blood were monitored at baseline, and then once weekly, using real-time, quantitative PCR (the unit is copies/mL). When 2 consecutive levels of CMV DNA > 500 copies/mL or EBV DNA > 1000 copies/mL were detected in the peripheral blood, CMV or EBV viremia was defined.

### 2.3. Treatments and Outcomes

Antiviral therapy included intravenous injection of ganciclovir (GCV, Cymevene, Roche Pharma Ltd., Basel, Switzerland, 3 or 6 mg/kg, q12 h × 2–3 weeks for without-induction therapy and 5 mg/kg/d × 1 week for maintenance therapy). Before intravitreal injection of GCV (IVG), aqueous humor was extracted. The aqueous CMV DNA load was determined by QNAT, and the aqueous interleukin-8 (IL-8) level was measured with flow cytometry, using a cytometric bead array. When the aqueous CMV DNA load decreased <2 orders of magnitude, or when the IL-8 level decreased <1 order of magnitude within the first 2 weeks after treatment initiation, intravitreal foscarnet (2.4 mg/0.2 mL) was administrated, along with ganciclovir [10].

The treatment termination was defined as negative CMV DNA (<1.0 × 10^3^ copies/mL) or negative IL-8 in the aqueous humor (<30 pg/mL). Recurrence was defined when the number of CMVR lesions, or the size of the preexisting lesions after treatment termination, increased [10]. The visual outcome was classified as improvement, stabilization, and deterioration. Improvement was defined as BCVA that increased by two or more lines. The definition of stabilization was that the changed BCVA was less than two lines. Deterioration was defined as a decreased BCVA of more than two lines [11]. The occurrence of retinal detachment was also recorded.

### 2.4. Statistical Analysis

All analyses were performed using Stata (version 18.0; College Station, TX 77845, USA) and R software version 3.6.1 (R Foundation for Statistical Computing, Vienna, Austria). For patient characteristics and the data of clinical examinations, we applied descriptive methods, with standard summary statistics including the mean (S.D., standard deviation), median, interquartile range (IQR), and proportions. To compare the differences between categorical variables, the Chi-square test with Fisher’s exact correction was used. The Student’s *t*-test and Mann–Whitney test were used for continuous variables between groups.

We used the least absolute shrinkage and selection operator (LASSO) and logistic regression model to select the most useful prognostic factors for recurrence and retinal detachment. R software version 3.6.1 and the “glmnet” package were used to perform the LASSO regression analysis [12]. Subsequently, variables identified by the LASSO regression analysis were incorporated into logistic regression models.

Univariate and multivariate analysis, conducted with a forward stepwise method, was performed to determine the visual prognostic factors, based on a logistic regression model. In the multivariate logistic regression analysis, variables with *p*  <  0.025 in the univariate analysis were included. All *p*-values were considered statistically significant if < 0.05.

## 3. Results

### 3.1. Patient Characteristics

The patient characteristics are summarized in Table 1. A total of 54 CMVR patients (84 eyes) were included in the study. Twenty-four patients suffered unilateral CMV retinitis and thirty were involved bilaterally. The mean age upon diagnosis of CMVR was 28.6 ± 11.5 years old, and thirty-two of them were males (59.3%). Primary hematological disorders included acute lymphoblastic leukemia (fourteen patients), acute myeloid leukemia (twenty-seven patients), lymphoma (four patients) and aplastic anemia, myelodysplastic syndrome (seven patients), and others. The donors (fifty related and four unrelated) of allo-HSCT included five HLA-matched and forty-nine HLA-mismatched donors. The proportions of poor graft function and GVHD were 25.9% and 70.4%. Nine (16.7%) of the CMVR patients had other organs involved in their CMV infection.

The demographic characteristics of patients with bilateral involvement or unilateral involvement were compared in Appendix A. A higher proportion of ABO match was found in bilaterally involved patients (*p* = 0.043).

### 3.2. Clinical Characteristics and Treatment Outcomes of CMVR

Table 2 presented the clinical characteristics and treatment outcomes of CMVR eyes. Patients were followed up for a median of 333 (IQR, 200–640) days. The median duration from HSCT to CMVR diagnosis was 148 (IQR, 119–196) days. During follow-up, 22 (26.2%) eyes developed CMV DNAemia at the time of CMVR diagnosis, and the median cumulative duration of CMV DNA viremia from HSCT to CMVR diagnosis was 39 (IQR, 24–52) days. The mean peak blood CMV DNA before CMVR diagnosis was 4.53 ± 0.55 log10 (copies/mL). There were 35 (41.7%) eyes with EBV DNAemia at the time of CMVR diagnosis. Thirty-one (36.9%) eyes had macula involved. Additionally, the median lesion area presented as DA was 67.5 (IQR, 36.4–161.4). Retinal lesions involving one, two, three, and four quadrants were found in fifteen (17.9%), twenty-five (29.8%), thirteen (15.5%), and thirty-one (36.9%) eyes, respectively.

Eighteen (21.4%) and sixty-six (78.6%) eyes were treated with without-induction and induction-maintenance therapy, and there were sixty-seven (79.8) and seventeen (20.2) eyes with low doses and high doses of ganciclovir, respectively. Among them, 29 (34.5%) eyes received an intravitreal foscarnet sodium injection. The median number of IVs was 6 (range, 4–8). The mean aqueous CMV DNA load at baseline was 4.22 ± 1.09 log10 (copies/mL).

When the treatment ceased, thirty-four (40.5%) eyes were negative for CMV DNA only, nine (10.7%) were negative for IL-8 only, twenty-four (28.6%) were negative for both, and seventeen (20.2%) eyes were clinically cured. Finally, improved or stabilized visual outcomes were found in 62 (73.8%) eyes, while 22 eyes (26.2%) suffered from deterioration. Eight (9.5%) eyes had rhegmatogenous retinal detachment and thirteen (15.5%) eyes suffered from recurrence.

### 3.3. Prognostic Factors for Visual Outcomes

The influence of clinical characteristics and treatment on the visual outcomes were shown in Table 3. Improved or stabilized visual outcomes were found in 22 eyes, while 62 eyes suffered from deterioration. There were more males in the group of deterioration. Patients who had a greater lesion area (*p* = 0.005) exhibited deteriorated VA. In addition, patients with deteriorated VA received more IVs (*p* = 0.014). A higher aqueous CMV DNA load (*p* = 0.025) and IL-8 (*p* = 0.024) at baseline indicated poorer visual outcomes. Factors with a *p*-value of less than 0.025 were incorporated in the univariate analysis.

In the univariate analysis (Table 4), males (OR 0.253, *p* = 0.024, 95% CI, 0.077–0.834), a greater lesion area (OR 0.989, *p* = 0.002, 95% CI, 0.982–0.996), and more IVs (OR 0.853, *p* = 0.022, 95% CI, 0.745–0.977) were found to be the significant prognostic factors for visual deterioration; conversely, in the multivariate analysis, lesion area was the only factor independently associated with the visual outcome (OR 0.989, *p* = 0.002, 95% CI, 0.981–0.996). Additionally, there were more eyes with retinal detachment (*p* = 0.002) in the group of deteriorated visual outcome, as shown in Appendix A. No significant difference in the rates of recurrence was found between the improved or stabilized group and the deteriorated group (Appendix A).

### 3.4. Prognostic Factors for Retinal Detachment and Recurrence

To predict the occurrence of retinal detachment, 29 variables (Appendix A) were included in the LASSO regression. After LASSO regression selection (Figure 1), seven variables remained significant predictors of retinal detachment. Inclusion of these seven variables in a logistic regression model resulted in one variable (aqueous CMV DNA load at baseline) that independently, in a statistically significant manner, predicted the retinal detachment (OR 5.087, *p* = 0.026, 95%CI 1.217–21.262), as indicated in Table 5.

Thirteen eyes (15.5%) suffered from recurrence, and there was no significant difference in the rates of recurrence between bilateral involved and unilateral involved patients (20% vs. 16.7%, *p* = 0.754). The 29 variables were also incorporated in the LASSO regression for the recurrence, as presented in Appendix A. After the LASSO regression selection (Figure 2), five variables remained significant predictors. In the logistic regression, macula involvement and IVs were found to be the independently prognostic factors for recurrence. A higher proportion of macula involvement (OR 5.322, *p* = 0.032, 95%CI 1.159–24.428) and more IVs (OR 1.263, *p* = 0.008, 95%CI 1.064–1.500) may indicate higher rates of recurrence (Table 6).

## 4. Discussion

CMVR is a vision-threatening disease in the immunocompromised population if it is not evaluated in a timely manner and adequately treated [2]. There have been very few related studies to elucidate the prognostic indicators of CMVR after HSCT. In this study, we retrospectively summarized the clinical and laboratory features of CMVR in 54 patients (84 eyes), as well as performed a comprehensive analysis of the prognostic factors for visual outcome, retinal detachment, and recurrence of CMVR. It is of note that many systemic and HSCT-associated factors were also included.

We found that gender had an impact on the visual outcome. In Wang’s study, B-cell reconstitution was delayed in males [13]. In the late posttransplant period, male recipients had lower B-cells than female recipients, which may be associated with the hormonal effects on B-cell development and function. Males were the negative factor for visual outcome in CMVR after allo-HSCT. Gender may affect the course of CMVR and subsequently impact visual prognosis by influencing the process of immune reconstitution after allo-HSCT. In addition, the lesion area was independently associated with the visual outcome. Larger lesions indicated a poorer immune status, which may impact the final VA. On the other hand, most retinitis lesions resolved with an area of retinal atrophy, and CMVR patients with a greater area tended to have permanent visual impairment and a lesser chance of vision improvement. Mo et al. proposed that a high intraocular CMV load and mismatched receipts may be potential negative factors affecting visual outcomes in CMVR following allo-HSCT [11], which was not found in our study. The different scale of our studies may explain the different results. In Wong’s study, macula involvement may reflect a poor visual outcome in CMVR patients without HIV infection [14]. Tao et al. also found that macula involvement and the initial visual acuity are significantly associated with visual prognosis in HIV-negative CMVR patients [15]. In our study, macula involvement was not the independent factor for visual outcome, although the proportions of macula involvement were significantly different between the two groups with improved or deteriorated vision. These findings prompted the importance of paying attention to patients with male gender, a larger lesion area, or macula involvement. Common risk factors of a poor visual outcome in CMVR patients included poor initial VA, retinal detachment, and a high plasma CMV titer, in Ping’s research [16]. Similarly, there were more eyes with retinal detachment in the group of deteriorated visual outcomes in our study. For CMVR patients with a larger lesion area at baseline, more aggressive treatment was needed to improve the visual outcome. Once retinal detachment developed, vitroretinal surgery could be conducted, depending on the patient’s situation. In addition to this, more intensive follow-up should be considered for CMVR patients with retinal detachment.

Our study also found that a higher aqueous CMV DNA load at baseline was related to higher rates of retinal detachment. Jabs et al. previously suggested that a poor immune status and a large area of CMVR were risk factors for retinal detachment in affected patients. CMVR patients with high intraocular CMV loads may have poorer immune status and will experience greater challenges in antiviral therapy, leading to worse prognosis [11]. The visual prognosis of CMV-related retinal detachment was poor, even if anatomical reattachment is successfully achieved. Retinal detachment is a late complication in CMVR eyes complicated with HSCT, which usually resulted from delayed diagnosis and severe inflammation due to impaired immune reconstruction. In our study, patients were scheduled to be followed up regularly after HSCT; furthermore, CMVR could be diagnosed once developed, and treatment was also given immediately upon CMVR diagnosis. Despite active treatment, retinal detachment occurred during the follow-up, which suggests that the severity of the disease was the main cause for retinal detachment. Therefore, to prevent retinal detachment in CMVR patients, regular examinations, education about the possibility of retinal detachment, and prophylactic peripheral laser treatment are necessary for patients with a high aqueous CMV DNA load.

The level of incompetency of the immune system highly impacts the risk of CMV recurrence, presented as an impairment of T-cell immunity, such as the presence and function of CMV-specific cytotoxic T lymphocytes [17,18,19,20,21]. Macula involvement and more IVs indicated higher rates of recurrence in CMVR after allo-HSCT in our study. We speculated that macula involvement and more IVs may reflect a more severe status of CMVR, which could result in a higher risk of recurrence. Similarly, in previous studies, foveal involvement was associated with poor visual outcomes in CMVR following transplantation, since most macular lesions resolved with retinal atrophy [14,22]. The extent of fundus lesions and the presence of vitreous haze were associated with recurrence [15]. Ting et al. reported that the HIV group had a significantly lower recurrence rate than the non-HIV group [16]. Furthermore, Boonsopon explains that those receiving systemic immunosuppression prior to CMVR presentation had a lower risk of CMVR recurrence [23]. These studies implied that immune status was crucial for the recurrence. Intense follow-up density is necessary for observing CMVR recurrence after HSCT. Patients should be closely followed up after their condition is stable. Some factors were not evaluated in our study due to the retrospective design. Larger prospective controlled trials will be carried out to determine the factors that could affect recurrence.

There were several limitations in this study. First, its retrospective nature cannot be neglected, which may cause some bias. Prospective cohort studies should be conducted in the future. Second, although it was the largest study yet to elucidate the prognostic factors of CMVR following all-HSCT, the number of patients who developed retinal detachment or CMVR recurrence was relatively small, and the prognostic analysis for retinal detachment and recurrence may be challenged. Overall, the sample size was small. Future studies with larger cohorts and multicenter collaborations could enhance the understanding of CMVR in the context of HSCT and improve the applicability of the findings to a broader clinical practice. In addition, we only incorporated patients with hematological diseases, and it is difficult to generalize the results of this study to other immunosuppressed patients.

In conclusion, this is the largest study to comprehensively investigate both ocular and systemic factors that may affect the clinical outcome of CMVR and establish a prognostic prediction model for CMVR patients after allo-HSCT. The lesion area was independently negatively associated with the visual outcome, and a higher aqueous CMV DNA load at baseline was the adverse prognostic factor for retinal detachment. Patients with macula involvement and more IVs had significantly higher rates of recurrence, which suggests the need for more intensive treatment. Thus, patients who are suspected to have poor prognosis should have routine ophthalmic screening after HSCT to ensure that CMVR is diagnosed as early as possible and treated appropriately.

## Figures and Tables

**Figure 1 biomedicines-13-00242-f001:**
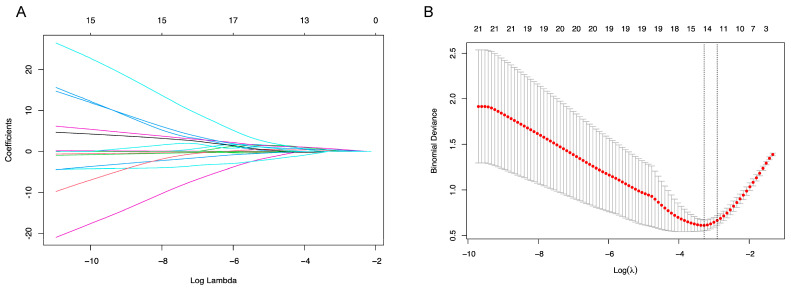
Feature selection using the least absolute shrinkage and selection operator (LASSO) binary logistic regression model for retinal detachment. (**A**) LASSO coefficient profiles of the 29 baseline features. (**B**) Tuning parameter (λ) selection in the LASSO model used 10-fold cross-validation via minimum criteria.

**Figure 2 biomedicines-13-00242-f002:**
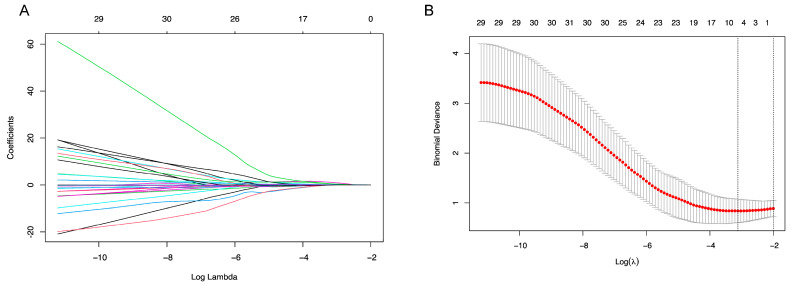
Feature selection using the least absolute shrinkage and selection operator (LASSO) binary logistic regression model for recurrence. (**A**) LASSO coefficient profiles of the 29 baseline features. (**B**) Tuning parameter (λ) selection in the LASSO model used 10-fold cross-validation via minimum criteria.

**Table 1 biomedicines-13-00242-t001:** Characteristics of patients with CMVR after HSCT.

Parameters		Total (n = 54)
Involved eye, n (%)		
	Both eyes	30 (55.6)
	Right eye	10 (18.5)
	Left eye	14 (25.9)
Age, mean ± SD		28.6 ± 11.5
Female:male		22:32
Hematological disease, n (%)		
	Acute lymphoblastic leukemia	14 (25.9)
	Acute myeloid leukemia	27 (50)
	Lymphoma	4 (7.4)
	Aplastic anemia and myelodysplastic syndrome	7 (13.0)
	Other	2 (3.7)
HLA match: mismatch		5:49
Donor related: unrelated		50:4
ABO match: mismatch		24:30
Poor graft function, n (%)		14 (25.9)
GVHD, n (%)		38 (70.4)
Other organ involvement with CMV infection, n (%)		9 (16.7)
Mean time of neutrophil engraftment, mean ± SD		14.5 ± 3.3
PTLD, n (%)		7 (13.0)
Duration of lymphopenia, median (IQR)		64 (47, 113)

CMVR, cytomegalovirus retinitis; HSCT, hematopoietic stem cell transplantation; SD, standard deviation; HLA, human leukocyte antigen; GVHD, graft-versus-host disease; IQR, interquartile range; PTLD, post-lymphoproliferative disease.

**Table 2 biomedicines-13-00242-t002:** Clinical parameters and treatment outcomes for eyes with CMVR after HSCT.

Parameters		Eyes (n = 84)
Clinical		
Duration from HSCT to CMVR diagnosis, median (IQR)		148 (119, 196)
Follow-up period from HSCT to last visit, median (IQR)		333 (200, 640)
Cumulative duration of CMV DNA viremia before CMVR diagnosis, median (IQR)		39 (24, 52)
Peak blood CMV DNA load before CMVR diagnosis, log10 (copies/mL), mean ± SD		4.53 ± 0.55
CMV DNAemia at time of CMVR diagnosis, n (%)		22 (26.2)
EBV DNAemia at CMVR diagnosis, n (%)		35 (41.7)
Macula involvement, n (%)		31 (36.9)
Lesion area, DA, median (IQR)		67.5 (36.4, 161.4)
Involved quadrants, n (%)		
	1 quadrant	15 (17.9)
	2 quadrants	25 (29.8)
	3 quadrants	13 (15.5)
	4 quadrants	31 (36.9)
Active inflammation in anterior segment, n (%)		15 (17.9)
Treatment		
Therapy, n (%)		
	Without-induction	18 (21.4)
	Induction-maintenance	66 (78.6)
Dose of ganciclovir, n (%)		
	3 mg	67 (79.8)
	6 mg	17 (20.2)
Combined with foscarnet sodium, n (%)		29 (34.5)
IVs, median (range)		6 (4, 8)
logMAR BCVA, median (IQR)		
	at the initial visit	0.349 (0.097, 1)
	at the last visit	0.349 (0.097, 1.561)
IOP, mmHg, median (IQR)		
	at the initial visit	14 (12, 17)
	at the last visit	12.5 (11, 13)
Aqueous CMV DNA load at baseline, log10 (copies/mL), mean ± SD		4.22 ± 1.09
Aqueous IL-8 level at baseline, pg/mL, median (IQR)		170.4 (49.9, 434.9)
Status at treatment termination, n (%)		
	CMV DNA-negative	34 (40.5)
	IL-8 negative	9 (10.7)
	CMV DNA and IL-8 both negative	24 (28.6)
	Clinical cure	17 (20.2)
Vision outcome, n (%)		
	improvement or stabilization	62 (73.8)
	deterioration	22 (26.2)
Retinal detachment, n (%)		8 (9.5)
Recurrence, n (%)		13 (15.5)

CMVR, cytomegalovirus retinitis; HSCT, hematopoietic stem cell transplantation; IQR, interquartile range; SD, standard deviation; EBV, Epstein–Barr virus; DA, disc area; IVs, intravitreal injections; BCVA, best-corrected visual acuity; IOP, intraocular pressure; IL, interleukin.

**Table 3 biomedicines-13-00242-t003:** Clinical characteristics and treatment of CMVR patients with different visual outcomes.

Parameters		Improvement or Stabilization (n = 22)	Deterioration (n = 62)	*p*-Value
Age, mean ± SD		29.1 ± 12.0	26.7 ± 10.5	0.409
Female: male		29:33	4:18	0.018 **
Hematological disease, n (%)				0.148
Acute lymphoblastic leukemia		7 (31.8)	16 (25.8)	
Acute myeloid leukemia		10 (45.5)	31 (50)	
Lymphoma		1 (4.5)	6 (9.7)	
Aplastic anemia and myelodysplastic syndrome		1 (4.5)	8 (12.9)	
Other		3 (13.6)	1 (1.6)	
HLA match: mismatch		6:56	3:19	0.606
Donor related: unrelated		6:56	2:20	0.936
ABO match: mismatch		31:31	13:9	0.463
Poor graft function, n (%)		14 (22.6)	9 (40.9)	0.098
GVHD, n (%)		43 (69.4)	14 (63.6)	0.622
Other organ involvement with CMV infection, n (%)		10 (16.1)	5 (22.7)	0.488
Mean time of neutrophil engraftment, mean ± SD		14.7 ± 3.9	14.5 ± 3.3	0.775
PTLD, n (%)		8 (12.9)	5 (22.7)	0.274
Duration of lymphopenia, median (IQR)		79.6 ± 39.6	83.5 ± 59.3	0.745
Duration from HSCT to CMVR diagnosis, median (IQR)		155 (119, 230)	145.5 (129, 173)	0.986
Follow-up period from HSCT to last visit, median (IQR)		319 (197, 709)	337.5 (216, 640)	0.891
Cumulative duration of CMV DNA viremia before CMVR diagnosis, median (IQR)		37 (24, 51)	48 (30, 78)	0.144
Peak blood CMV DNA before CMVR diagnosis, log10 (copies/mL), mean ± SD		4.45 ± 0.56	4.75 ± 0.47	0.04 *
CMV DNAemia at time of CMVR diagnosis, n (%)		18 (29.0)	4 (18.2)	0.32
EBV DNAemia at CMVR diagnosis, n (%)		29 (50.9)	6 (31.6)	0.144
Macula involvement, n (%)		18 (36)	13 (62)	0.045 *
Lesion area, DA, median (IQR)		59.5 (32.2, 113.3)	159.5 (49.1, 220.4)	0.005 **
Active inflammation in anterior segment, n (%)		12 (19.4)	3 (13.6)	0.547
Therapy, n (%)				0.101
	Without-induction	16 (25.8)	2 (9.1)	
	Induction-maintenance	46 (74.2)	20 (90.9)	
Dose of ganciclovir, n (%)				0.37
	3 mg	48 (77.4)	19 (86.4)	
	6 mg	14 (22.6)	3 (13.6)	
Combined with foscarnet sodium		19 (30.7)	10 (45.5)	0.209
IVs, mean ± SD		6.0 ± 2.8	8.2 ± 5.0	0.014 **
Aqueous CMV DNA load at baseline, log10 (copies/mL), mean ± SD		3.90 (3.12, 4.86)	4.88 (3.83, 5.54)	0.025 **
Aqueous IL-8 level at baseline, pg/mL, median (IQR)		139.6 (35.7, 355.5)	338.7 (155, 564.5)	0.024 **

* *p* < 0.05; ** *p* < 0.025. CMVR, cytomegalovirus retinitis; HLA, human leukocyte antigen; GVHD, graft-versus-host disease; HSCT, hematopoietic stem cell transplantation; IQR, interquartile range; SD, standard deviation; PTLD, post-lymphoproliferative disease; EBV, Epstein–Barr virus; DA, disc area; IVs, intravitreal injections; IL, interleukin.

**Table 4 biomedicines-13-00242-t004:** Prognostic factors affecting the visual outcome of CMVR occurring after HSCT using logistic regression.

Parameters	Univariate			Multivariate		
	*p*-Value	95%CI	OR	*p*-Value	95%CI	OR
Female: male	0.024 **	0.077–0.834	0.253	0.05	0.058–0.999	0.240
Aqueous CMV DNA load at baseline	0.064	0.388–1.027	0.632			
Aqueous IL-8 level at baseline	0.802	0.999–1.000	1			
Macula involvement	0.048	0.121–0.992	0.346			
Lesion area	0.002 **	0.982–0.996	0.989	0.017 **	0.981–0.998	0.989
IVs	0.022 **	0.745–0.977	0.853	0.097	0.754–1.023	0.879

** *p* < 0.025. CMVR, cytomegalovirus retinitis; HSCT, hematopoietic stem cell transplantation; CI, confidence interval; OR, odds ratio; CMV, cytomegalovirus; IL-8, interleukin-8; IVs, intravitreal injections.

**Table 5 biomedicines-13-00242-t005:** Logistic regression of prognostic factors for retinal detachment of CMVR after HSCT.

Parameters	Univariate			Multivariate		
	*p*-Value	95%CI	OR	*p*-Value	95%CI	OR
Age	0.052	0.854–1.000	0.924			
Hematological disease	0.221	0.800–2.624	1.450			
Poor graft function	0.031 **	1.168–24.7	5.37	0.648	0.135–24.891	1.834
Therapy	NA	NA	NA			
Aqueous CMV DNA load at baseline	0.005 **	1.610–14.094	4.764	0.026 **	1.217–21.262	5.087
Lesion area	0.003 **	1.005–1.025	1.015	0.674	0.988–1.018	1.003
LogMAR at last visit	0.002 **	1.744–11.158	4.411	0.133	0.742–0.595	2.668

** *p* < 0.025. CMVR, cytomegalovirus retinitis; HSCT, hematopoietic stem cell transplantation; CI, confidence interval; OR, odds ratio; CMV, cytomegalovirus.

**Table 6 biomedicines-13-00242-t006:** Logistic regression of prognostic factors for recurrence of CMVR after HSCT.

Parameters	Univariate			Multivariate		
	*p*-Value	95%CI	OR	*p*-Value	95%CI	OR
Hematological disease	0.188	0.849–2.300	1.397			
Therapy	NA	NA	NA			
IVs	0.003 **	1.085–1.496	1.274	0.008 **	1.064–1.500	1.263
Aqueous CMV DNA load at baseline	0.304	0.429–1.302	0.747			
Macula involvement	0.024 **	1.233–20.649	5.045	0.032 **	1.159–24.428	5.322

** *p* < 0.025. CMVR, cytomegalovirus retinitis; HSCT, hematopoietic stem cell transplantation; CI, confidence interval; OR, odds ratio; CMV, cytomegalovirus; IVs, intravitreal injections; NA, not applicable.

## Data Availability

The data supporting the findings are not publicly available due to privacy or ethical restrictions. The data are available upon reasonable request from the corresponding author.

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
