# Peer review of "Clinical Characteristics and Prognostic Factors Affecting Clinical Outcomes in Cytomegalovirus Retinitis Following Allogeneic Hematopoietic Stem Cell Transplantation"

_biomedicines, 2025, doi:10.3390/biomedicines13010242_

Round 1
Reviewer 1 Report
Comments and Suggestions for Authors
The manuscript presents a retrospective cohort study investigating the clinical characteristics and prognostic factors associated with cytomegalovirus retinitis (CMVR) in patients who underwent allogeneic hematopoietic stem cell transplantation (allo-HSCT). The study is well-structured and adheres to ethical guidelines, with appropriate statistical analyses.
Comments:
1- The introduction could provide more background on CMVR and its significance in the context of allo-HSCT.
2- The manuscript could benefit from a more detailed discussion of the implications of the findings for clinical practice
3- There is a lack of discussion regarding the limitations of the study, particularly concerning the retrospective design
4- The discussion lacks a comparison with existing literature, which would help contextualize the findings
5- While the methods are sound and the study is well-conducted, the sample size of 54 CMVR patients (84 eyes) may be considered relatively small, which could limit the generalizability of the results. The sample size presents a limitation that should be acknowledged. Future studies with larger cohorts and possibly multicenter collaborations could enhance the understanding of CMVR in the context of allo-HSCT and improve the applicability of the findings to broader clinical practice.
6- The reference list is extensive and relevant. However, ensure that all citations are formatted consistently according to the journal's guidelines.
7- A list of abbreviations should be added to the manuscript
Author Response
Reviewer 1
The manuscript presents a retrospective cohort study investigating the clinical characteristics and prognostic factors associated with cytomegalovirus retinitis (CMVR) in patients who underwent allogeneic hematopoietic stem cell transplantation (allo-HSCT). The study is well-structured and adheres to ethical guidelines, with appropriate statistical analyses.
Comment 1: The introduction could provide more background on CMVR and its significance in the context of allo-HSCT.
Response 1: Thank you for your valuable suggestion. We have added more background on CMVR and its significance in the context of allo-HSCT as follows, according to your comment. (Page 1, line 37-Page 2, line 50).
“Cytomegalovirus retinitis (CMVR) is a severe retinal disease typically observed in immunocompromised patients which could lead to poor visual outcome, such as allogeneic hematopoietic stem cell transplantation (allo-HSCT) patients[1,2]. It manifested variably accordingto different degrees of CMV-specific leukocytes deficit and immunosuppression[3]. Major retinal lesions include irregular white necrotic foci or edema surrounded by granular infiltrates, spreading centrifugally along the vessels, with or without focal retinal hemorrhage[4-6].
Allo-HSCT is a well-established treatment for several hematological diseases. However, the broadening of indication, new preconditioning regimens, more aggressive treatment of graft-versus-host disease (GVHD) and the prolonged survivor rate after HSCT unfavorably results in more opportunistic infections including CMVR[6,7], of which the incidence ranges from 0.2 to 5.6%[6-9]. Latent CMV activation or exogenous CMV entering the retina through the blood-eye barrier could cause CMVR. The CMV serostatus of the donor or recipient, antecedent CMV reactivation, delayed lymphocyte engraftment, and occurrence of GVHD have been proposed as risk factors for CMVR after allo-HSCT[9]. It can cause various blurred vision, dark shadow floating, visual field loss, vision loss and other severe lesions. Without in-time diagnosis and effective treatment, patients may suffer progressive retinal necrosis and optic atrophy, and eventually permanent vision loss.”
Comment 2: The manuscript could benefit from a more detailed discussion of the implications of the findings for clinical practice.
Response 2: We appreciated for your insightful comment. In this study, we found that lesion area was independently negatively associated with the visual outcome, and higher aqueous CMV DNA load at baseline was the adverse prognostic factor for retinal detachment. Patients with macula involvement and more IVs, which suggest the need of more intensive treatment, had significantly higher rates of recurrence. We briefly discussed about the implications of those findings in the original manuscript, and have added more discussion in Page 12, line 261-283 and line 295-296.
“Those finding prompted the importance of paying attention to patients with male gender, larger area of lesion or macula involvement. Risk factors of poor visual outcome in common CMVR included poor initial VA, retinal detachment, and a high plasma CMV titer in Ping’s research[16]. Similarly, there were more eyes with retinal detachment in the group of deteriorated visual outcome in our study. For CMVR patients with larger lesion area at baseline, more aggressive treatment was needed to improve visual outcome. Once retinal detachment developed, vitroretinal surgery could be given depend on the patient’s situation. Besides, more intensive follow-up should be considered for CMVR patients with retinal detachment.
Our study also found that higher aqueous CMV DNA load at baseline was related with higher rates of retinal detachment. Jabs et al previously suggested that a poor immune status and a large area of CMVR were risk factors for retinal detachment in affected patients. CMVR patients with high intraocular CMV loads may have poorer immune status and will experience greater challenges in antiviral therapy, leading to worse prognosis[11]. The visual prognosis of CMV-related retinal detachment was poor even if anatomical reattachment is successfully achieved. Retinal detachment is a late complication in CMVR eyes complicated with HSCT, which usually resulted from delayed diagnosis and severe inflammation due to impaired immune reconstruction. In our study, patients were scheduled to be followed up regularly after HSCT, CMVR could be diagnosed once developed and treatment was also given immediately upon CMVR diagnosis. Despite active treatment, retinal detachment occurred during the follow-up, which suggest the severity of the disease was the main cause for retinal detachment. Therefore, to prevent retinal detachment in CMVR patients, regular examinations, education about the possibility of retinal detachment, and prophylactic peripheral laser treatment is necessary for patients with high aqueous CMV DNA load.
Intense follow-up density is necessary for observing CMVR recurrence after HSCT. Patients should be closely followed up after their condition is stable. Some factors were not evaluated in our study for the retrospective design. Larger prospective controlled trials will be carried out to determine these factors that could affect recurrence.”
Comment 3: There is a lack of discussion regarding the limitations of the study, particularly concerning the retrospective design.
Response 3: Many thanks for the valuable comment. We mentioned the limitation of retrospective design in the original manuscript. However, we agreed with you that more discussion about limitations should be elucidated, and we have revised the part of limitation in Page 13, line 299-line 308.
“There were several limitations in this study. First, its retrospective nature cannot be neglected, which may cause some bias. Prospective cohort studies should be conducted in the future. Second, although it was the largest study yet to elucidate the prognostic factors of CMVR following all-HSCT, the number of patients who developed retinal detachment or CMVR recurrence was relatively small, and the prognostic analysis for retinal detachment and recurrence may be challenged. Overall,the sample size was small. Future studies with larger cohorts and multicenter collaborations could enhance the understanding of CMVR in the context of HSCT and improve the applicability of the findings to broader clinical practice. In addition, we only incorpo-rated patients with hematological diseases, and it is difficult to generalize the results of this study to other immunosuppressed patients.”
Comment 4: The discussion lacks a comparison with existing literature, which would help contextualize the findings.
Response 4: Thank you for the great suggestion. In the original manuscript, the comparison with existing literature was not enough to contextualize the findings. As you suggested, we have added more comparison with existing literature, as follows in Page 12, line 254-261, and Page 12, line 288-295.
“Mo et al proposed that high intra-ocular CMV load and mismatched receipts may be potential negative factors affecting visual outcomes in CMVR following allo-HSCT[11], which was not found in our study. Different scale of our studies may explain for the different results. In Wong’s study, macula involvement may reflect poor visual outcome in CMVR without HIV infection[14]. Tao et al also found that macula involvement and initial visual acuity significantly associate with visual prognosis in HIV-negative CMVR[15]. In our study, macula involvement was not the independent factor for visual outcome, although the proportions of macula involvement were significantly different between the two groups with improved or deteriorated vision.
Similarly in previous studies, foveal involvement was associated with poor visual outcomes in CMVR following transplantation, since most macular lesions resolved with retinal atrophy[14,22]. The extent of fundus lesions, and the presence of vitreous haze were associated with the recurrence[15]. Ting et al reported that the HIV group had significantly lower recurrence rate than the non-HIV group[16]. And Boonsopon mentioned that receiving systemic immunosuppression prior to CMVR presentation had a lower risk of CMVR recurrence[23]. Those studies implied that immune status was crucial for the recurrence.”
Comment 5: While the methods are sound and the study is well-conducted, the sample size of 54 CMVR patients (84 eyes) may be considered relatively small, which could limit the generalizability of the results. The sample size presents a limitation that should be acknowledged. Future studies with larger cohorts and possibly multicenter collaborations could enhance the understanding of CMVR in the context of allo-HSCT and improve the applicability of the findings to broader clinical practice.
Response 5: We appreciated your great comment and suggestions. The limitation of sample size was briefly mentioned in the original manuscript. As you suggested, we have improved our expression and acknowledged the limitation of sample size in detail, as shown in Page 13, line 300-308.
“Second, although it was the largest study yet to elucidate the prognostic factors of CMVR following all-HSCT, the number of patients who developed retinal detachment or CMVR recurrence was relatively small, and the prognostic analysis for retinal detachment and recurrence may be challenged. Overall, the sample size was small. Future studies with larger cohorts and multicenter collaborations could enhance the understanding of CMVR in the context of HSCT and improve the applicability of the findings to broader clinical practice. In addition, we only incorporated patients with hematological diseases, and it is difficult to generalize the results of this study to other immunosuppressed patients.”
Comment 6: The reference list is extensive and relevant. However, ensure that all citations are formatted consistently according to the journal's guidelines.
Response 6: We appreciated your comment and have used the MDPI.ens references style according to the journal's guidelines in the revised manuscript (Page 19, line 389-Page 20, line 447).
Comment 7: A list of abbreviations should be added to the manuscript.
Response 7: Thank you for your reminder. We totally agreed with this comment. The list of abbreviations was supplemented and shown in the Page 14, line 344-Page 15, line 366.
Reviewer 2 Report
Comments and Suggestions for Authors
This study provides valuable insights into the clinical characteristics and prognostic factors affecting visual outcomes, retinal detachment, and recurrence in patients with cytomegalovirus retinitis (CMVR) following allogeneic hematopoietic stem cell transplantation (allo-HSCT). The retrospective analysis of 54 patients (84 eyes) highlights that ocular factors, such as lesion size, macula involvement, and the number of intravitreal injections, are critical predictors of visual deterioration and recurrence, with baseline aqueous CMV DNA load being a significant predictor of retinal detachment. The study’s use of logistic and LASSO regression models strengthens the reliability of the findings. However, further studies with larger sample sizes and longer follow-up periods would be beneficial to validate these findings and explore additional risk factors. I have some concerns:
1. To predict the occurrence of retinal detachment, 29 variables (Supplementary Table S2) were included in the LASSO regression. Theoretically, the sample size for LASSO regression should be at least greater than the number of features. Generally, the sample size should be at least 10 times (or more) the number of features. How does the author approach this issue?
2. It is necessary to provide the reference for the "glmnet" package.
Author Response
Reviewer 2
This study provides valuable insights into the clinical characteristics and prognostic factors affecting visual outcomes, retinal detachment, and recurrence in patients with cytomegalovirus retinitis (CMVR) following allogeneic hematopoietic stem cell transplantation (allo-HSCT). The retrospective analysis of 54 patients (84 eyes) highlights that ocular factors, such as lesion size, macula involvement, and the number of intravitreal injections, are critical predictors of visual deterioration and recurrence, with baseline aqueous CMV DNA load being a significant predictor of retinal detachment. The study’s use of logistic and LASSO regression models strengthens the reliability of the findings. However, further studies with larger sample sizes and longer follow-up periods would be beneficial to validate these findings and explore additional risk factors. I have some concerns:
Comment 1: To predict the occurrence of retinal detachment, 29 variables (Supplementary Table S2) were included in the LASSO regression. Theoretically, the sample size for LASSO regression should be at least greater than the number of features. Generally, the sample size should be at least 10 times (or more) the number of features. How does the author approach this issue?
Response 1: We appreciated for your comment. The sample size should be at least 10 times (or more) the number of features, mainly in the logistic or Cox regression. The commonly used variable selection methods include subset selection method and coefficient compression method [1]. The former constructs equations by combining all possible independent variable combinations with the dependent variable, and selects the best fitting result from them. In this process, even small changes in variables can affect the selection result, and the computational cost is particularly large. The number of equations constructed is exponentially related to the number of independent variables, which is not suitable for situations with a large number of independent variables [2]; The latter, including LASSO, can overcome the disadvantages of unstable subset selection results and high computational complexity, as well as requirement for large sample size.
Lasso regression was first proposed by Robert Tibshirani, and is now widely used in prediction models. It is recommended to first consider using Lasso regression for variable screening when fitting models with too many variables and a small sample size. When the number of events is small compared with the number of regression coefficients, model overfitting can be a serious problem. An overfitted model tends to demonstrate poor predictive accuracy when applied to new data. LASSO may alleviate overfitting by shrinking the regression coefficients towards zero [3]. Therefore, LASSO was used in many studies with limited sample size. We had in-depth discussions with statisticians in our hospital about your concern. And we will carry out studies with larger sample size in the future to better determine our results. The discussion about limitation of sample size was added in Page 13, line 300-Page 13, line 306.
“Second, although it was the largest study yet to elucidate the prognostic factors of CMVR following all-HSCT, the number of patients who developed retinal detachment or CMVR recurrence was relatively small, and the prognostic analysis for retinal detachment and recurrence may be challenged. Overall,the sample size was small. Future studies with larger cohorts and multicenter collaborations could enhance the understanding of CMVR in the context of allo-HSCT and improve the applicability of the findings to broader clinical practice.”
References:
[1] ChongIG, JunCH. Performance of some variable selection methods when multicollinearity is present[J]. Chem Intell Lab Sys, 2005, 78( 1-2): 103- 112. DOI: 10.1016/j.chemolab.2004.12.011.
[2] 奚丽婧, 郭昭艳, 杨雪珂, 等. LASSO及其拓展方法在回归分析变量筛选中的应用[J]. 中华预防医学杂志, 2023, 57(1): 107-111. DOI: 10.3760/cma.j.cn112150-20220117-00063
[3] Pavlou M, Ambler G, Seaman S, De Iorio M, Omar RZ. Review and evaluation of penalised regression methods for risk prediction in low-dimensional data with few events. Stat Med. 2016 Mar 30;35(7):1159-77. doi: 10.1002/sim.6782. Epub 2015 Oct 29. PMID: 26514699; PMCID: PMC4982098.
Comment 2: It is necessary to provide the reference for the "glmnet" package.
Response 2: Thank you for your valuable suggestion. We have added the reference for the "glmnet" package in Page 19 line 418, which was as follows:
Engebretsen, S.; Bohlin, J. Statistical predictions with glmnet. Clin Epigenetics 2019, 11, 123, doi:10.1186/s13148-019-0730-1.